# Effectiveness of the Quadrivalent HPV Vaccine in Preventing Anal ≥ HSILs in a Spanish Population of HIV+ MSM Aged > 26 Years

**DOI:** 10.3390/v13020144

**Published:** 2021-01-20

**Authors:** Carmen Hidalgo-Tenorio, Juan Pasquau, Mohamed Omar-Mohamed, Antonio Sampedro, Miguel A. López-Ruz, Javier López Hidalgo, Jessica Ramírez-Taboada

**Affiliations:** 1Infectious Disease Department, Hospital Universitario Virgen de las Nieves Granada, 18014 Granada, Spain; jpasquau@gmail.com (J.P.); mangel.lopez.ruz.sspa@juntadeandalucia.es (M.A.L.-R.); jessicart2003@gmail.com (J.R.-T.); 2Infectious Disease Unit, Complejo Hospitalario Jaén, 23007 Jaén, Spain; omarampa@gmail.com; 3Microbiology Department, Hospital Universitario Virgen de las Nieves, 18014 Granada, Spain; antonioj.sampedro.sspa@juntadeandalucia.es; 4Pathology Department, Hospital Universitario Virgen de las Nieves, 18014 Granada, Spain; javierl.lopez.sspa@juntadeandalucia.es

**Keywords:** quadrivalent HPV vaccine, anal high-grade squamous intraepithelial lesion (HSIL), human immunodeficiency virus (HIV), men who have sex with men (MSM), anal cancer

## Abstract

Anal squamous cell carcinoma is the most frequent virus-related non-AIDS-defining neoplasia among HIV-infected individuals, especially MSM. The objectives of this study were to analyze the effectiveness of the quadrivalent HPV (qHPV) vaccine to prevent anal ≥ high-grade squamous intraepithelial lesions (≥HSILs), external ano-genital lesions (EAGLs), and infection by qHPV vaccine genotypes in HIV+ MSM, and to study the immunogenicity of the vaccine and risk factors for ≥ HSILs. This study is nested within a randomized, double-blind, placebo-controlled trial of the qHPV vaccine, which enrolled participants between May 2012 and May 2014, with a 48-month follow-up. A vaccine or placebo was administered at 0, 2, and 6 months, and vaccine antibody titers were evaluated at 7, 12, 24, 36, and 48 months. Data were gathered at 12, 24, 36, and 48 months on sexual habits, CD4/CD8 cell/counts, HIV viral load, and the results of cytology (Thin Prep^®^ Pap Test), HPV PCR genotyping (Linear Array HPV Genotyping Test), and high-resolution anoscopy (Zeiss 150 fc^©^ colposcope). The study included 129 patients (mean age of 38.8 years, 40 [31%] with a history of AIDS, 119 [92.2%] receiving ART, and 4 [3.3%] with virological failure), 66 (51.2%) in vaccine arm and 63 (48.4%) in placebo arm. The vaccine and placebo groups did not differ in ≥ HSILs (14.1 vs. 13.1%, respectively, *p* = 0.98) or EAGL (11.1 vs. 6.8%, *p* = 0.4) rates during follow-up; however, a protective effect against HPV 6 was observed during the first year of follow-up in the vaccine *versus* placebo group (7.5% vs. 23.4%; *p* = 0.047). A between-arm difference (*p* = 0.0001) in antibodies against qHPV vaccine genotypes was observed at 7 months (76.9% in vaccine arm vs. 30.2% in placebo arm), 12 months (68.1% vs. 26.5%), 24 months (75% vs. 32.5%), 36 months (90% vs. 24.4%), and 48 months (87.2% vs. 30%). Finally, the factor associated with the risk of anal ≥ HSIL onset during the four-year follow-up was the receipt of the last dose of the vaccine less than 6 months earlier in comparison to those vaccinated for a longer period (82.4% vs. 17.6% (OR 0.869 [95% CI, 0.825–0.917]). Vaccine and placebo arms did not significantly differ in ≥ HSIL or EAGL rates or in protection against infection by HPV genotype vaccine except for HPV6 at 12 months after the first dose. A long-lasting immune response was observed in almost all the vaccinated men. The main protective factor against ≥ HSIL was to have completed the vaccination regimen more than 6 months earlier.

## 1. Introduction

The incidence of anal squamous cell carcinoma (ASCC) is elevated in HIV-infected patients, especially in men who have sex with men (MSM) and in women with cervical disease [1], being much higher than in the general population [2]. ASCC progresses rapidly in the absence of medical treatment, and some authors have proposed its early diagnosis by the cytological analysis of anal swabs as a cost-effective strategy [3]. Persistent infection by human papillomavirus (HPV) is a risk factor for ASCC and anal ≥ high-grade squamous intraepithelial lesions (HSILs) [4,5], and some authors described the development of HSIL in up to 70% of patients with persistent infection by genotypes 16/18 for a median of 20 months range, 6–36 months [6].

Approaches to ASCC prevention include programs for screening, diagnosis, treatment, and follow-up of HSILs [4,7]; the utilization of condoms to avoid HPV infection [8]; and the administration of HPV vaccine [9]. 

Various clinical trials have been published on the immunogenicity and/or effectiveness of the quadrivalent HPV (qHPV) vaccine in the anal-genital area of HIV seronegative MSM. A comparison between MSM and men who have sex with women (MSW) found that the latter were at lower risk of external genital lesions (Intention-to Treat Population: 0.51/100 person-year at risk in MSM vs. 1.27 person-year at risk in MSM; and Per-Protocol Population: 0.08/100 person-year at risk in MSW vs. 0.42/100 person-year at risk in MSM) [10], and no differences were found between MSM and MSW in the formation of antibodies related to the vaccine (at month 36: antibody against HPV-6 in 89.5% of MSW vs. 80% of MSM; against HPV-11 in 94.3 vs. 89.1%; against HPV-16 in 98.3 vs. 93.9%; and against HPV-18 in 57.3 vs. 53.6%, respectively) [11]. Another non-randomized, open, one-arm trial in HIV seronegative MSM reported an effectiveness rate of 50% against HSIL onset [12]. Various clinical trials have studied the immunogenicity and safety of the qHPV vaccine in HIV+ adult MSM. In one of these, virtually all patients developed antibodies against the qHPV vaccine genotypes [13]. However, the latest trial in adult HIV+ men and women was interrupted due to a lack of effectiveness against HSIL onset [14].

In 2017, our group published the first results of this clinical trial on the immunogenicity and safety of the qHPV vaccine in HIV+ MSM, reporting that 70% of the vaccinated patients had antibodies to HPV at seven months after the first dose [15]. The present study reports on the effectiveness of the vaccine in this population after a 48-month follow-up. The objectives were: to evaluate the effectiveness of the qHPV vaccine against anal ≥ HSILs, external ano-genital lesions (EAGLs), and infection with HPV vaccine genotypes in HIV+MSM; and to determine the immunogenicity of the vaccine over four years and risk factors for ≥ HSILs.

## 2. Materials and Methods

Detailed information on the patients and methods used in our randomized, double-blind, placebo-controlled trial of the qHPV vaccine was previously published [15]. In summary, the sample comprised HIV-positive MSM ≥ 18 years of age not simultaneously infected by the four qHPV genotypes or both genotypes 16 and 18, who had normal high-resolution anoscopy (HRA) or an anal biopsy finding of condyloma and/or anal low-grade squamous intraepithelial lesion (LSIL) alone. Participants were enrolled between 15 May 2012 and 15 May 2014. The trial (ISRCTN14732216. DOI 10.1186/ISRCTN14732216) complied with the protocols of the Spanish Drugs and Health Products Agency [Agencia Española del Medicamento y Productos Sanitarios; AEMPS] and was approved by the local ethics committee. All data were coded to ensure anonymity, and all participants signed their fully informed consent. The protocol is available at http://www.isrctn.com/ISRCTN14732216.

In brief, using a methodology previously described in detail [15], data were collected at baseline and at 12, 24, 36, and 48 months on clinical-epidemiological variables and blood analysis results, including the HIV viral load (VL). Antibodies against the four qHPV vaccine genotypes were determined at month 7 (after the 3rd dose) and at 12, 24, 36, and 48 months.

The analyses were performed in the hospital microbiology department by a single microbiologist (A.S.). The HPVG ELISA commercial kit (DIA.PRO, Milano, Italy), which measures antibodies against the major capsid protein (L1) of HPV types 6, 11, 16, and 18 was applied according to the manufacturer’s instructions. Blood samples were centrifuged and frozen at −20 °C until their analysis. The results are expressed as positive or negative. 

The HRA, PCR, and cytology techniques employed are described in detail elsewhere [15]. HPV genotypes 16, 18, 26, 31, 33, 35, 39, 45, 51–53, 56, 58, 59, 66, 68, 73, and 82 were considered high risk (HR-HPV), and genotypes 6, 11, 34, 40, 42–44, 54, 55, 57, 61, 70–72, 81, 83, 84 and 89 low-risk (LR-HPV). HPV-18 virus species was classified as genotypes 39, 45, 59, 68 and HPV-16 species as genotypes 31, 33, 35, 52, 58, 67 [16]. Lesions were categorized according to the Bethesda cytology classification [17] as atypical squamous cells of undetermined significance (ASCUS), or atypical squamous cells—cannot rule out HSIL—(ASC-H). We adopted the histology classification of the Squamous Terminology (LAST) Standardization Project for HPV, categorizing lesions as LSIL (AIN1/condyloma), HSIL (AIN2, AIN3), or invasive carcinoma (ASCC) [18]. 

The participants were randomly assigned (1:1 ratio) to receive an 0.5 mL injection of qHPV vaccine (Gardasil^©^; Merck Research Laboratories, West Point, PA, USA) or placebo at day 1, month 2, and month 6 [15]. 

### 2.1. Sample Size

The sample size was calculated to estimate a reduction of ≥50% of HSIL after qHPV vaccination with a statistical power of 80% and level of significance of 5%, based on previous findings of pathological anal biopsy in 67.2% of HIV+ MSM, with 29.8% of these having HSIL (≥AIN2) and 74.2% showing infection by high-risk genotypes [19]. A sample size of 29 patients per arm was estimated, which was increased to 60 per arm to yield a statistical power of 98% [15]. 

### 2.2. Randomization

The participants were randomly assigned to the experimental or placebo group using a randomization sequence created with Epidat software (Epidat 4.2, 2016. Consellería de Sanidade, Xunta de Galicia, España; Organización Panamericana de la salud (OPS-OMS); Universidad CES, Medellín, Colombia). Patients and researchers were blinded to the allocation of participants. 

### 2.3. Statistical Analysis

SPSS (version 19) and Stata Statistical Software Release 12 packages were used for statistical analyses. Means, standard deviation, medians, and percentiles were calculated for quantitative variables and absolute and relative frequencies for qualitative variables. The prevalence of HPV and ≥ HSIL in anal mucosa was determined with 95% confidence interval (CI). The Kolmogorov–Smirnov test was applied to check the distribution of variables. Factors associated with infection by HR-HPV genotypes were analyzed using the Student *t*-test for quantitative variables with normal distribution and the Mann–Whitney test for those with non-normal distribution, and using the Pearson’s chi-square test or Fisher test, as appropriate, for qualitative variables; the FDR correction was applied for multiple comparisons [20]. 

Multivariate Poisson regression analysis was then performed, including variables significant in bivariate analyses and others deemed clinically relevant. The variables were entered in the model by forward stepwise selection, setting the *p*-value of 0.05 or 0.10 to enter the model. *p* < 0.05 was considered statistically significant in all tests. 

## 3. Results

### 3.1. Patients Enrolled in Clinical Trial

Out of 162 HIV+ MSM patients undergoing screening from 15 May 2012 through 15 May 2014, 30 (18.5%) did not meet inclusion criteria and 3 (1.85%) withdrew their consent. The 129 patients in the final study sample were randomly assigned to the vaccine arm (*n* = 66, 51.2%) or placebo arm (*n* = 63, 48.4%). Among the 66 participants in the vaccine arm, 64 (96.9%) received all three doses of the vaccine and the remaining 2 participants only two doses, due to death from liver cirrhosis at 6 months post-enrolment and relocation to another city, respectively. The follow-up was completed by 110 (85.3%) participants, with a median of 48 months (P25: 36-P75 48), but not by 19 (14.7%), due to non-AIDS-defining disease in two cases (10.5%; lung cancer in 1 and decompensated liver cirrhosis in 1), with the remaining 17 (89.4%) being lost to the follow up (Figure 1). 

The mean age of the participants was 38.8 years, 69 (53.5%) had completed university studies, their sexual life had started a median of 19 years earlier, 40 (31%) had a history of AIDS, 119 (92.2%) were receiving antiretroviral treatment, and only 4 (3.3%) had virological failure. Table 1 lists the results for remaining variables, showing that patients in each arm were similar in age, social strata, educational level, sexual habits, consumption of toxic substances, other infections, and HIV-related data (virological and immunological status). 

With respect to anal infection by HPV genotype, 90 (73.8%) patients were infected with high-risk genotypes, 73 (59.8%) with low-risk genotypes, and 59 (48.4%) with both types of virus; 30 (24.6%) were infected with genotype 16, 19 (15.6%) with genotype 6, 15 (12.6%) with genotype 11, and 9 (7.4%) with genotype 18. Cytology results were normal in 53 (41.1%) patients and revealed LSIL in 60 (46.5%), HSIL in 5 (3.9%), and ASC-US in 11 (8.5%). Anal biopsy evidenced LSIL/AIN1 in 62 patients (48.1%) and was normal in 67 (51.9%). There was no between-group difference in genotype infection or in cytology or histological results (Table 2). 

The prevalence of HSILs and ASCC during follow-up of the global cohort was, respectively, 10.5% (13/124) and 0.8% (1/124) at 12 months, 0.95% (1/105) and 0% at 24 months, 1.02% (1/98) and 0% at 36 months, and 1.08% (1/93) and 0% at 48 months. The difference in ≥ HSIL prevalence between 12 months (11.3% [14/124]) and 48 months (1.07% [1/93]) was statistically significant (*p* < 0.0001). There was also a decrease in the incidence of HSILs and ASCC, which was 104.8 × 1000 p-year and 8.07 × 1000 p-year, respectively, at 12 months of follow-up, 62.8 × 1000 p-year and 4.48 × 1000 p-year at 24 months, 48.54 × 1000 p-year and 3.4 × 1000 p-year at 36 months; and 41.03 × 1000 p-year and 2.56 × 1000 p-year at 48 months. The difference in the incidence of ≥ HSIL between 12 months and 48 months was also statistically significant (*p* < 0.0001). The only case of invasive perianal cancer was diagnosed in a patient who had been vaccinated 6 months earlier; it debuted with a hardened perianal lesion and a metastatic cervical lymph node in which high-risk genotype 33 was detected. Despite receiving chemotherapy and radiotherapy, he died at 12 months. 

### 3.2. Efficacy of qHPV Vaccine

The ≥ HSIL rate did not significantly differ between the vaccinated and placebo groups during the 48-month follow-up (9 [14.1%] vs. 8 [13.1%], respectively, *p* = 0.87) (Table 3), finding no difference at 12 months (7/63 [11.1%] vs. placebo 7/61 [11.5%], respectively; *p* = 1), 24 months (0 [0%] vs. 1/50 [2%], *p* = 0.472), 36 months (1/53 [1.9%] vs. 0%, *p* = 1), or 48 months (1/50 [2%] vs. 0 [0%], *p* = 1). 

During the 48-month follow-up, vaccine and placebo arms did not differ in the prevention of EAGLs/condylomas (EAGL (7 patients [11.1%] vs. 4 patients [6.8%], respectively, *p* = 0.4) (Table 3). 

No significant between-arm difference was found in the acquisition of qHPV genotypes at 12 months with the exception of HPV-6, which was observed in 4 (7.5%) of the vaccine group vs. 11 (23.4%) of the placebo group (*p* = 0.047); the remaining results are displayed in Table 3. 

### 3.3. Immunogenicity of qHPV Vaccine

Antibodies against qHPV vaccine genotypes were detected in 76.9% (50/65) of the vaccine group vs. 30.2% (19/63) of the placebo group at 7 months (*p* < 0.0001), in 68.1% (32/47) vs. 26.5% (13/50) at 12 months (*p* < 0.0001), in 75% (36/48) vs. 32.5% (13/40) at 24 months (*p* = 0.0001); in 90% (45/50) vs. 24.4% (10/41) at 36 months (*p* < 0.0001); and in 87.2% (41/47) vs. 30% (12/40) at 48 months (*p* < 0.0001). The antibodies remained stable throughout the follow up.

### 3.4. Factors Related to ≥ HSIL Onset

The sole factor significantly associated with the risk of anal ≥ HSIL onset during the four-year follow-up was the receipt of the last dose of the vaccine less than 6 months earlier in comparison to those vaccinated for a longer period (82.4% vs. 17.6% (OR 0.869 [95% CI, 0.825–0.917]). Results for the remaining study variables are exhibited in Table 4. 

## 4. Discussion

The patients in this cohort were aged >26 years, had a very good virological/immunological status, and were under antiretroviral treatment, with 15.6, 12.6, 24.6, and 7.4% of them being previously infected with HPV genotypes 6, 11, 16, and 18, respectively. Around half of them had normal anal mucosa and around half had LISLs, while more than one-third were infected with subspecies of genotype 16 and around one-third with subspecies of genotype 18. Oncogenic virus 16 is the most frequently isolated HPV genotype in anal mucosa and has been related to EAGLs in both sexes [21,22]. In particular, HIV positivity is one of the main risk factors for infection with high-risk HPV genotypes in MSM [23]. 

No statistically significant differences between vaccine and placebo arms were found in the onset of ≥ HSILs or condylomas or in the acquisition of qHPV genotypes during the follow-up period, except for a lesser acquisition of genotype 6 at 12 months after the first dose in the vaccinated patients. 

The HPV vaccine is recommended for individuals aged between 12 and 26 years in most countries, including Spain, based on expectations of a higher effectiveness in a non-virus-exposed population and of a superior immune response in younger people [24]. However, it has also proven effective in adult women with previous exposure to the qHPV vaccine genotypes [25]. Moreover, the MAM trial found that the immune response of adult males aged 27 to 45 years was comparable to that of younger men and that the vaccine was effective against EAGLs and persistent HPV infection in men aged over 26 years [26]. However, a trial on the effectiveness of the qHPV vaccine in seropositive adults was interrupted because there was no reduction in anal HSIL onset related to the vaccine genotypes, although there was a decrease in the rate of infection by these genotypes; this failure may be in part explained by the inclusion of adult HIV patients of both sexes in the trial [14]. The vaccine was found to reduce the development of EAGLs and anal HSILs in HIV seronegative heterosexual men and MSM [10], with a slightly lesser reduction in the latter [12]. The above data suggest that the vaccine should be offered to seropositive MSM of any age who are engaged in an active sexual life.

At one month after completing the vaccination regimen, around 8 out of 10 vaccinated patients had total antibodies against the vaccine genotypes in comparison to three out of ten of those receiving placebo; furthermore, this difference was not only maintained during the follow-up but increased, with the percentage in the vaccine arm reaching around 90%. These outcomes confirm the long-lasting immunity conferred by the qHPV vaccine in seropositive MSM, in agreement with previous observations [12,26]. One study of HIV+ MSM found that 98% developed antibodies against the four vaccine genotypes [13], while another described a similar long-term immune response between seropositive and HIV seronegative men, although antibody levels were lower in the former [27]. At any rate, the clinical relevance of a low antibody concentration is unknown, and no concentration cutoff has been published that correlates with the vaccine’s effectiveness against external lesions or anal, cervical, vulvar, or vaginal intraepithelial neoplasia [1,14]. 

The main factor related to ≥ HSIL onset in our patient cohort was receipt of the last vaccine dose 6 months earlier, with only isolated cases being observed at 24, 38, and 48 months after completing the vaccine regimen. This may be explained by a slightly delayed immune response in these older adults, with antibodies being detected in a large proportion of vaccinated patients at 12 months after the first dose and in an even greater percentage at 24 months, remaining close to 90% until the end of the 48-month follow-up. 

The limitations of the trial include the rigorous eligibility criteria, leading to the exclusion of screened patients for the presence of anal ≥ HSIL (14.2%) or for anal mucosal infection with both genotypes 16 and 18 (4.5%). Further research is therefore necessary to evaluate the usefulness of the vaccine in patients with these characteristics. 

## 5. Conclusions

This cohort of adult HIV+ MSM evidenced a reduction in anal ≥ HSILs and EAGLs over the follow-up period, with no significant difference between those receiving the qHPV vaccine and those receiving placebo. According to these findings, it does not appear recommendable to administer the qHPV vaccine to HIV+ MSM over the age of 26 years. However, the receipt of the vaccine offered protection against the acquisition of genotype 6 during the first 12 months of follow-up. The main protective factor against ≥ HSIL onset in these HIV+ MSM was to have completed the vaccination regimen more than 6 months earlier.

## Figures and Tables

**Figure 1 viruses-13-00144-f001:**
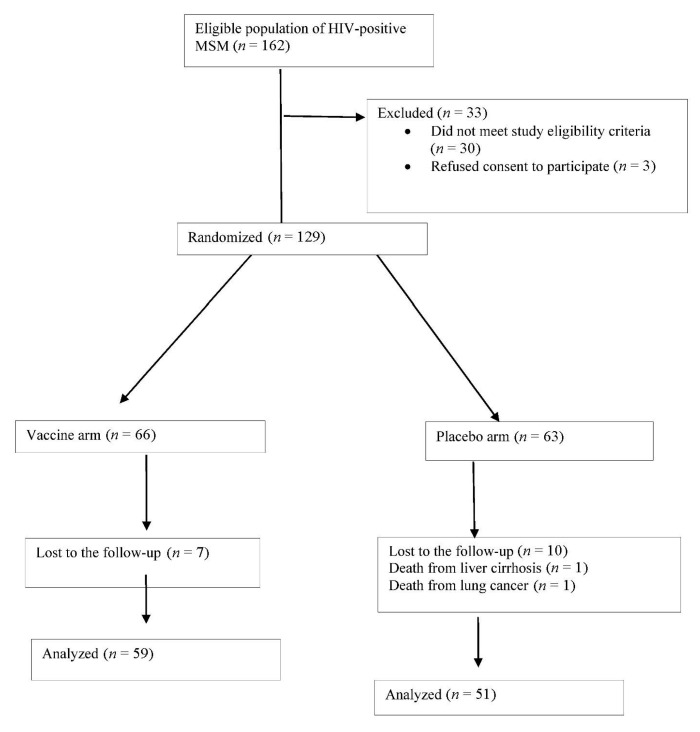
Flow of patients through the study.

**Table 1 viruses-13-00144-t001:** Baseline demographics of HIV-positive men who have sex with men (MSM) enrolled in the clinical trial.

Variables	Total Cohort of HIV+ MSM (*n* = 129)	HIV+ MSM Vaccine (*n* = 66)	HIV+ MSM Placebo (*n* = 63)	**p*
Age, years; mean (±SD)	38.8 (±10.4)	37.3 (±10.6)	40.5 (±10.02)	0.082
Spanish nationality, *n* (%)	123 (95.3)	63 (95.5)	60 (95.2)	0.2
University education, *n* (%)	69 (53.5)	34 (51.5)	35 (55.5)	0.56
Partners in previous 12 months; median (IQR)	1 (1–3.75)	1 (1–3)	1 (1–5)	0.8
Life-time partners; *n*, median (IQR)	72.5 (20.5–300)	50 (20–300)	100 (45–350)	0.041 *
Years of sexual activity; median (IQR)	19 (10–27)	17 (9–24)	21 (13–27)	0.025 *
Condom use, *n* (%)	100 (77.5	53 (80.3)	47 (74.6)	0.4
Perianal/genital condylomas at screening, *n* (%)	40 (31)	20 (30.3)	20 (31.7)	0.86
History of condylomas, *n* (%)	34 (26.4)	19 (28.8)	15 (23.8)	0.52
Duration of HIV; mean months (IQR)	67.5 (31–123.5)	58 (26–120)	77 (37–138)	0.2
History of AIDS; *n* (%)	40 (31)	19 (28.8)	21 (33.3)	0.58
CD4 mean nadir; cells/µL (±SD)	335,12 (±210.9)	336 (±227.3)	334.2 (±193.7)	0.96
CD4 mean; cells/µL (±SD)	721.9 (±258.8)	733 (±252.7)	710.4 (±266.6)	0.62
CD8 mean; cells/µL (±SD)	996.2 (±422.04)	999.9 (±463.6)	992.2 (±374.9)	0.98
VL of HIV log10; copies/mL (±SD)	3.72 (±4.84)	3.76 (±4.5)	3.67 (±4.46)	0.8
VL < 50 copies/mL, *n* (%)	106 (82.2)	53 (80.3)	53 (84.1)	0.57
Virological failure, *n* (%)	4 (3.3)	1 (1.5)	3 (4.8)	0.29
ART, *n* (%)	119 (92.2)	60 (90.9)	59 (93.7)	0.745
Median duration of ART; months (IQR)	45 (17–101.25)	42 (17–86)	43 (17–129)	0.42
Number of lines of ART, median (IQR)	1 (1–2)	1 (1–2)	1 (1–2)	0.56
Syphilis treated, *n* (%)	28 (21.7)	16 (24.2)	12 (19.1)	0.47
Other STD, *n* (%)	23 (17.8)	11 (16.6)	12 (19.1)	0.72
Latent tuberculosis treated, *n* (%)	15 (11.6)	5 (7.6)	10 (15.9)	0.14
HCV, *n* (%)	4 (3.1)	2 (3)	2 (3.2)	1
HBV, *n* (%)	2 (1.6)	2 (3)	0 (0)	0.49
Smoking, packs/year, median (IQR)	1.9 (0–12.5)	0.2 (0–1)	6,5 (0–18)	0.008 *
Ex-smoking, *n* (%)	23 (17.8)	10 (15)	13 (20.6)	0.42
Ex-IVDA, *n* (%)	1 (0.8)	1 (1.5)	0 (0)	0.42
Alcohol (standard units of alcohol; SUA)	0.16 (0–1)	0 (0–1)	0.4 (0–1.4)	0.15

**p*: *p*-value; *p* < 0.05. ART: antiretroviral therapy. VL: viral load; Ex-IDVA: ex-injection drug venous addiction. * FDR correction for multiple comparisons showed that difference was not statistically significant.

**Table 2 viruses-13-00144-t002:** Baseline human papillomavirus (HPV) PCR, cytology, and high-resolution anoscopy (HRA) * results in the study population.

Variable	Total Cohort of HIV+ MSM (*n* = 122)	HIV+ MSM Vaccine (*n* = 65)	HIV+ MSM Placebo (*n* = 57)	**p*
**PCR of HPV, *n* (%)**				
LR-HPV	73 (59.8)	39 (60)	34 (59.6)	0.97
HR-HPV	90 (73.8)	46 (70.8)	44 (77.2)	0.42
LR and HR HPV, *n* (%)	59 (48.4)	28 (43.1)	31 (54.4)	0.21
**Number of HR-HPVs (IQR)**	1 (0–3)	1 (0–2)	2 (0–3)	0.22
**Number of LR-HPVs (IQR)**	1 (0–2)	1 (0–2)	1 (0–2)	0.94
**Genotypes, *n* (%)**				
HPV6	19 (15.6)	11 (16.9)	8 (14)	0.66
HPV11	15 (12.6)	8 (12.3)	7 (12.3)	0.99
HPV16	30 (24.6)	15 (23.1)	15 (26.3)	0.68
HPV16 species	46 (37.8)	22 (33.3)	24 (38.1)	0.49
HPV18	9 (7.4)	4 (6.2)	5 (8.8)	0.58
HPV18 species	40 (32.8)	18 (27.3)	22 (34.9)	0.35
**Cytology, *n* (%)**				
Normal	53 (41.1)	26 (39.4)	27 (42.9)	0.69
LSIL	60 (46.5)	34 (51.5)	26 (41.3)	0.24
HSIL	5 (3.9)	1 (1.5)	4 (6.3)	0.21
ASCUS	11 (8.5)	5 (7.6)	6 (9.5)	0.69
	**Total Cohort of**	**HIV+ MSM**	**HIV+ MSM**	
	**HIV+ MSM**	**Vaccine**	**Placebo**	
**HRA, *n* (%)**	***n* = 129**	***n* = 66**	***n* = 63**	
Normal	67 (51.9)	33 (50)	29 (46)	0.65
LSIL(AIN1)	62 (48.1)	33 (50)	34 (54)	0.65

HRA: high resolution anoscopy. **p: p*-value; *p* < 0.05.

**Table 3 viruses-13-00144-t003:** Efficacy of qHPV vaccine.

Variable	HIV-MSM Vaccine (*n* = 65)	HIV-MSM Placebo (*n* = 63)	**p*
**HSIL**			
12 m–48 m, *n* (%)	9 (14.1)	8 (13.1)	0.87
12 m, *n* (%)	7/63 (11.1)	7/61 (11.4)	1
24 m, *n* (%)	0 (0)	1/50 (2)	0.47
36 m, *n* (%)	1/53 (1.9)	0 (0)	1
48 m, *n* (%)	1/50 (2)	0 (0)	1
**EAGL/Condyloma**			
12 m–48 m, *n* (%)	7 (11.1)	4 (6.8)	0.40.6
12 m, *n* (%)	3 (4.6)	1 (1.7)	0.7
24 m, *n* (%)	2/54 (3.7)	3/51 (5.9)	0.5
36 m, *n* (%)	2/52 (3.8)	0 (0)	1
48 m, *n* (%)	0 (0)	0 (0)	
**Acquisition HPV Genotypes 12 m**	***n* = 53**	***n* = 47**	
HPV 6, *n* (%)	4 (7.5)	11 (23.4)	0.047
HPV 11, *n* (%)	4 (7.5)	3 (6.3)	1
HPV 16, *n* (%)	12 (25.5)	10 (24.4)	1
HPV 18, *n* (%)	9 (15.8)	6 (11.8)	0.55
**Acquisition HPV Genotypes 24 m**	***n* = 48**	***n* = 43**	
HPV 6, *n* (%)	4 (8.3)	2 (4.6)	0.69
HPV 11, *n* (%)	6 (13)	1 (2.3)	0.11
HPV 16, *n* (%)	3 (9.4)	4 (9.3)	1
HPV 18, *n* (%)	3 (6.1)	1 (2.3)	0.62
**Acquisition HPV Genotypes 36 m**	***n* = 37**	***n* = 34**	
HPV 6, *n* (%)	1 (2.7)	3 (8.8)	0.34
HPV 11, *n* (%)	2 (5.4)	0 (0)	0.49
HPV 16, *n* (%)	4 (10.8)	1 (2.9)	0.36
HPV 18, *n* (%)	2 (5.4)	0 (0)	0.49
**Acquisition HPV Genotypes 48 m**	***n* = 30**	***n* = 27**	
HPV 6, *n* (%)	3 (10)	1 (3.7)	0.62
HPV 11, *n* (%)	0 (0)	0 (0)	1
HPV 16, *n* (%)	2 (6.7)	1 (3.7)	1
HPV 18, *n* (%)	2 (6.7)	0 (0)	0.49

**p* < 0.005.

**Table 4 viruses-13-00144-t004:** Bivariate and multivariate analyses of factors associated with ≥ HSILs.

Factors	HIV-MSM with ≥ HSIL (*n* = 17)	HIV-MSM without ≥ HSIL (*n* = 107)	**p*	RR; (95% CI)
q-HPV Vaccine, *n* (%)	9 (52.9)	55 (50.9)	0.877	
Ab q-HPV Vaccine, *n* (%)				
7 months	9 (52.9)	58 (54.2)	0.92	
12 months	8 (53.3)	37 (45.1)	0.56	
24 months	2 (100)	47 (54.7)	0.2	
36 months	1 (50)	54 (60.7)	1	
48 months	0	53 (61.6)	0.391	
Titers Ab 7 m, median (IQR)	2.21 (1.78–4.39)	3.46 (1.7–5.1)	0.41	
Titers Ab 12 m, median (IQR)	3.48 (2.26–3.65)	2.39 (0–4.53)	0.65	
Interval between first dose and HSIL onset, IQR	12 (12–12)	48 (48–48)	0.0001	0.869 (0.825–0.917)
Age; mean years (±SD)	39 (10.98)	38.54 (10.28)	0.62	
Median HR-HPV, (IQR)	1 (0–4)	1 (0–3)	0.454	
Median LR-HPV, (IQR)	1 (0–2)	1 (0–2)	0.625	
University education, *n* (%)	8 (47.1)	59 (54.6)	0.626	
Retired, *n* (%)	2 (11.8)	11 (10.2)	0.69	
Partners in previous 12 months; median (IQR)	1 (0.5–2.5)	1 (1–4)	0.394	
Life-time partners; median (IQR)	100 (15–325)	70 (22.75–300)	0.988	
Time since start of sexual activity (years); median (IQR)	21 (11–28)	18.5(10–27)	0.552	
Condom use, *n* (%)	13 (76.5)	84 (77.8)	1	
% condom use; median (IQR)	99 (5–100)	100 (52.5–100)	0.289	
Perianal/genital condyloma, *n* (%)	7 (41.2)	32 (29.6)	0.4	
History of condyloma, *n* (%)	2 (11.8)	31 (28.7)	0.235	
Smoking, *n* (%)	10 (58.8)	43 (38.9)	0.12	
Smoking, packs/year; median (IQR)	2.7 (0–22)	1.1 (0–12)	0.267	
Ex-smoker, *n* (%)	2 (11.8)	20 (18.5)	0.497	
Ex-IVDA, *n* (%)	0	1 (0.9)	1	
Alcohol, consumer *n* (%)	6 (35.3)	57 (52.8)	0.18	
Alcohol (SDU), median, IQR	0 (0–1.5)	0.2 (0–1)	0.397	
Duration of HIV; mean (IQR)	90 (52.5–214.5)	61.5 (29.25–123.5)	0.138	
CD4 mean nadir; cells/µL (±SD)	299.59 (261.1)	341.4 (205.9)	0.456	
Naïve	1 (5.9)	9 (8.3)	1	
VL of HIV, log10(± SD)	2.54 (3.12)	3.79 (4.51)	0.469	
VL < 50 copies/mL, *n* (%)	15 (88.2)	87 (80.6)	0.736	
CD4 mean; cells/µL, (± SD)	696.92 (215)	729.01 (268.4)	0.64	
CD8 mean; cells/µL (± SD)	1038.4 (273.4)	995.6 (446)	0.71	
CD4/CD8	0.74 (0.29)	1.4 (5.85)	0.634	
Previous AIDS diagnosis; *n* (%)	8 (47.1)	31 (28.7)	0.129	
Median duration of ART; months (IQR)	51 (26.5–74)	55 (23–111)	0.214	
Virological failure, *n* (%)	0	4 (4)	0.418	
Syphilis, *n* (%)	4 (23.5)	22 (20.4)	0.753	
Other STD, *n* (%)	2(11.8)	20 (18.5)	0.735	
HCV, *n* (%)	0	3	1	
HBV, *n* (%)	0	2 (1.9)	1	
AIN1 (1st HRA), *n* (%)	9 (52.9)	57 (52.8)	0.99	
HR-HPV baseline, *n* (%)	13 (76.5)	74 (73.3)	1	
LR-HPV baseline, *n* (%)	11 (64.7)	59 (58.4)	0.625	
HR and LR-HPV baseline, *n* (%)	10 (58.8)	47 (46.5)	0.35	
HPV at baseline visit				
HPV-6, *n* (%)	3 (17.6)	16 (15.8)	1	
HPV-11, *n* (%)	3 (17.6)	12 (11.9)	0.452	
HPV-16, *n* (%)	5 (29.4)	24 (23.8)	0.761	
HPV-18, *n* (%)	1 (5.9)	7 (6.9)	1	
HPV 12 months after first dose				
HPV-6, *n* (%)	4 (23.5)	22 (22.4)	1	
HPV-11, *n* (%)	2 (11.8)	13 (13.3)	1	
HPV-16, *n* (%)	7 (41.2)	37 (37.8)	0.789	
HPV-18, *n* (%)	5 (29.4)	11 (11.2)	0.06	

*p**: *p*-value; *p* < 0.05. q-HPV vaccine: quadrivalent human papillomavirus vaccine; Ab q-HPV Vaccine: antibody to q-HPV vaccine. HRA: high resolution anoscopy. SDU: standard drink unit.

## Data Availability

Registration and protocol: ISRCTN14732216; available at http://www.isrctn.com/ISRCTN14732216.

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
