# Peer review of "Effectiveness of the Quadrivalent HPV Vaccine in Preventing Anal ≥ HSILs in a Spanish Population of HIV+ MSM Aged > 26 Years"

_viruses, 2021, doi:10.3390/v13020144_

Round 1
Reviewer 1 Report
Major comments
P2, Abstract, Results, L7-10: It is not immediately possible to understand what the sentence “A between arm difference in antibody formation was observed …” means. Later in the manuscript it becomes clear that this is referring to Results, P8, section 3.3, “Immunogenicity of qHPV vaccine”. This section is quite confused. It is incorrect to make statements about the immunogenicity of the placebo. You can describe HPV seroconversions due to natural exposure in the placebo group. To make any statements about immunogenicity of the vaccine (or indeed new seroconversions in the placebo group) you will first need to exclude the individual subjects-genotypes who are either HPV 6, 11,16,18 DNA positive or seropositive at entry. Once you have done this you can then describe seroconversions and antibody stability in the two separate groups. The data need to be re-analysed. The paragraph (bottom P15 / top P16) in the discussion will need to be re-considered in the light of further analyses
P2, Abstract, Results, L10-13: Again, it is not immediately possible to understand what this sentence means. Later, it becomes clear this is referring to P8, section 3.4, although again this section is confusing. But it seems both sections are referring to the multivariate analyses presented in Table 3, in particular the line “Interval between first dose and HSIL onset”. There is a conceptual problem here. The interval between first dose and HSIL onset in HIV-MSM without ≥HSIL was 48 months, i.e. the total length of the study, and by definition none of them had ≥HSIL, so the calculation of an RR is meaningless / incorrect. As far as I can understand the data presented, these statements have to be omitted. This includes the last sentence in the Abstract, Conclusions. Again, the Discussion, P16, 2nd paragraph will need to be re-considered. I do not believe the proposal of a “delayed immune response”.
P4, L12-15: “Comparison between MSM and men who have sex with women (MSW) found that the latter were at lower risk of external genital lesions (0.08/100 person-year at risk in MSW vs. 0.42/100 person-year at risk in MSM) [9]”. The use of “were at lower risk” is mis-representing the data and needs correcting. VE was the same in MSM vs MSW (70.2 vs 63.7%, NS); the difference in EGL incidence is due to behaviour / exposure.
P4, L15-18: “and more frequently generated antibodies (Abs) against vaccine genotypes (at month 36: Ab to HPV-6 in 89.5% of MSW vs. 80% of MSM; to HPV-11 in 94.3 vs. 89.1%; to HPV-16 in 98.3 vs. 93.9%; and to HPV-18 in 57.3 vs. 53.6%, respectively) [10]”. Again, the use of “more frequently” is mis-representing the data, you should clarify that these differences are not significant. I quote directly from the publication “Though not statistically significant, a higher proportion of HM subjects seroconverted for vaccine HPV types than did MSM subjects (Table 4).”
P7, 3.1, last paragraph:
Statement 1, “the prevalence of HSILs and ASCC during the follow-up of the global cohort, was 10.5% (13/124) and 0.8% (1/124), respectively, at 12 months, in 0.95% (1/105) and 0% at 24 months, in 1.02% (1/98) and 0% at 36 months, and in 1.075% (1/93) and 0%, respectively, at 48 months.”
Statement 2, “There was also a decrease in the incidence of HSILs and ASCC, which was 104.8 x 1000 p-year and 806.45 x 100.000 p-year, respectively, at 12 months of follow-up, 62.8 x 1000 p-year and 448.4 x 100.000 p-year at 24 months, 48.54 x 1000 p-year and 323.64 x 100.000 p-year at 36 months; and 41.03 x 1000 p-year and 256.4 x 100.000 p-year at 48 months.”
The use of different denominators in statement 2, i.e. /1000 patient years, and then /100,000 patient years is very confusing. Please stick to one format for these rates. There is also confusion re use of prevalence (statement 1) and incidence (statement 2). You state there was only one case of invasive anal cancer at 6 months who died at 12 months. How come you say there continues to be incidence (i.e. new cases) of anal cancer at 24, 36 & 48 months? Must be a mistake. And then when you come to Table 3, Line ‘Interval between first dose and HSIL onset’, Column ‘HIV-MSM with ≥HSIL’, the value is 12 (12-12). This appears to be stating that there were no new cases of HSIL after 12 months. If so the incidence of HSIL at 24, 36 & 48 months is 0. Please reconsider and revise.
Minor comments
P4, L12: the use of “seronegative MSM” is confusing (this could refer to either HPV or HIV), you should state “HIV seronegative MSM”. Happens again on L18. Please search for seronegative and seropositive and correct throughout the manuscript.
P5, Methods: As far as I can tell, HRA was only performed at entry and not during subsequent follow up. It also seems that subsequent cytological findings of anal HSIL were not further investigated by HRA or treated. This should be clarified in the Methods.
P5, L15: You refer to “the HPVG ELISA commercial kit (DIA.PRO, Milano, Italy)”. You should clarify that this assay measures IgG antibodies against VLPs of the four genotypes.
P5, L30: What was the placebo? Please clarify.
P12-14, Table 3: much too big, take out all the individual genotype lines in “HPV at baseline visit” and “HPV 12 months after first dose”
Reviewer 2 Report
Abstract Results:
- Few typos in abstract (missing parentheses in results)
Introduction: How rapidly do HSIL progress – from initial HPV infection to HSIL? This could be helpful to know
Introduction: Question: Does anal squamous cell carcinoma progress rapidly? If so, can you cite research that support this? The research you cited supported the statement that early detection is vital.
Introduction: The De Polomandy article you cited also concluded that co-infection with HPV 16 and 18 was a risk factor for progression from infection to HSIL. Include that information. (I think it would further support your study).
Results: Efficacy of qHPV vaccine – Could this information be put in a Table/Chart? As written, this is hard to follow. It may be in Table 3. This is also hard to follow. Can authors find a way to have side-by-side reporting of infection with the various HPV types (6, 11, etc) from baseline to the end of the 48 month study timeframe?
The conclusion – Do you think it is worth vaccinating HIV+ MSM over age 18? Over the age 26? (I saw that the mean age of the study sample is 38)
Conclusion: Do you have any idea as to why the vaccine was not found to protect against HSIL/EAGL recurrence? Maybe men were already infected with these 2 types?
Conclusion: Would authors recommend this vaccine to HIV+ MSM? It created antibodies but did not impact HSIL and EAGLs.
Author Response
"Please see the attachment.

Round 2
Reviewer 1 Report
The authors have made satisfactory revisions.
Author Response
Dear Reviewer
The paper has been revised by an expert editor (Richard Davies, Cambridge University graduate with 30 years' experience translating/editing scientific papers). Typographic errors have been detected and amended alongside some improvements in punctuation. We attach the revised version and a document indicating all of the changes introduced.
We trust that our paper is now suitable for publication.
With thanks,
Sincerely yours,